# Fermented Soybean Meal Affects the Reproductive Performance and Oxidative Status of Sows, and the Growth of Piglets

**DOI:** 10.3390/ani11030597

**Published:** 2021-02-24

**Authors:** Wenli Luo, Xiaofeng Yin, Jianbo Yao, Jinlong Cheng, Jing Zhang, Weina Xu, Yuyun Mu, Jianxiong Xu

**Affiliations:** 1Shanghai Key Laboratory of Veterinary Biotechnology, School of Agriculture and Biology, Shanghai Jiao Tong University, Shanghai 200240, China; maple_rowe@sjtu.edu.cn (W.L.); digestion2020@163.com (X.Y.); zhangjing224@sjtu.edu.cn (J.Z.); xuweina@sjtu.edu.cn (W.X.); 2Division of Animal and Nutritional Sciences, West Virginia University Morgantown, Morgantown, WV 26506, USA; Jianbo.Yao@mail.wvu.edu; 3Jiangsu Unison Biotechnology Development Co., Ltd., Xuzhou 223831, China; chengjinlong310@163.com; 4Shanghai Xinnong Feed Co., Ltd., Shanghai 201613, China; Yuyun.Mu@novusint.com

**Keywords:** colostrum, growth factor, immunoglobulin, oxidative stress, serum

## Abstract

**Simple Summary:**

Fermentation of the cooked soybean meal increases the contents of isoflavone aglycones, provides soy proteins that are easier to absorb and eliminates trypsin inhibitors in soybean meal. In the present study, replacing soybean meal in the maternal diet with fermented soybean meal decreased the oxidative stress in the serum of sows and increased the average body weight of piglets on the 14th day and the 21st day after birth. We concluded that replacing the soybean meal in the maternal diet with the fermented soybean meal attenuates the oxidative stress status of the gestational and lactational sows, and enhances the average weight of their offspring.

**Abstract:**

This study aimed to investigate the effect of the fermented soybean meal on the reproductive performance, oxidative stress and colostrum composition of sows, and the growth performance of their progeny. A total of 44 sows were allotted to four dietary groups (*n* = 11/group). The dietary groups included the basal diet group (control) and the treatment groups in which soybean meal in the basal diet was replaced with 2%, 4%, and 6% fermented soybean meal, respectively. The experimental diets were fed to the sows from the 78th day of gestation to the 21st day of lactation. Replacing soybean meal in the basal maternal diet with the fermented soybean meal decreased the levels of malondialdehyde, cortisol, and 8-iso-prostaglandinF2α in the serum of sows and increased the average weight of piglets on the 14th day and the 21st day after birth. The activity of superoxide dismutase in the serum of sows was increased in the group with 4% fermented soybean meal on the 17th day of lactation. The levels of estrogen and growth factors in the serum of sows were enhanced in the group with 6% fermented soybean meal. In the colostrum, the levels of the protein and the immunoglobulin G were enhanced in the group with 4% fermented soybean meal. In conclusion, replacing the soybean meal in the basal maternal diet with the fermented soybean meal attenuates the oxidative stress status of the gestational and lactational sows, and enhances the average weight of their offspring.

## 1. Introduction

Soybean meal (SM) is one of the basal dietary components for sows as the main source of protein. SM contains a variety of biologically active ingredients, such as soybean isoflavones (SIFs), which include different types such as genistein and daidzein with a spectrum of biological activities. SIFs are excellent antioxidants, which can reduce lipid oxidation, inhibit the free radical formation, and stimulate anti-oxidative enzymes [1,2,3]. SIFs exist mostly as the forms of glycoside in soybean, but rarely as the forms of aglycone, which are absorbed easier by the intestinal wall [4,5]. Fermentation of the cooked soybeans increased the contents of isoflavone aglycones with high efficacy of biotransformation, while decreased the total contents of isoflavone glycosides by about 80% [6]. Our previous studies showed that the ratio of the SIFs as aglycone to the total SIFs in the fermented soybean meal (FSBM) can reach 64.56% [7]. Furthermore, fermentation of SM is able to provide soy proteins that are easier to absorb, because the process of fermentation eliminates trypsin inhibitors, increases the amount of protein in SM, and reduces the particle size of peptides in SM [8].

There are adequate antioxidant reserves to handle free radical production in normal physiological body status, but the status of oxidative stress increases, and the production of the antioxidant in the body circulation markedly decreases during late gestation and lactation period of sows [9,10]. When the equilibrium is broken during the gestation and lactation period, oxidative stress develops. This process may impact the reproductive performance of the sows and the development of their progeny [11]. The growth of new-born piglets almost completely depends on the nutrients from the colostrum and milk of sows [12]. Nutrients in the maternal diet affect the production and components of breast milk, so the nutrient requirement of sows is a key for feeding sows for producing high quality and sufficient milk to support the development of their progeny [13]. However, there is no report about the effect of the maternal diet with FSBM on the immunoglobulins in the colostrum of sow. In addition, previous reports showed that replacing SM with FSBM enhanced nutrient utilization efficiency in weaning piglets, modulated the response of inflammation and the activity of oxidant in nursing piglets, and alleviated stress responses in Holstein calves after weaning [14,15]. So far, there is no report about the effect of replacing SM with FSBM in maternal diets on the oxidative stress status of sows and the performance of their suckling offspring. Therefore, considering the potential benefits of FSBM, we hypothesized that replacing SM with FSBM in maternal diet could decrease the oxidative stress in sows, improve the contents of protein, fat, lactose, total solids (TS), solid not fat (SNF), and immunoglobulins in colostrum and further improve the performance of their offspring.

The present study was conducted to investigate the effect of replacing SM with FSBM in the maternal diets on anti-oxidative status, reproductive performance, colostrum composition of sows during gestation and lactation period, and the performance of their offspring.

## 2. Materials and Methods

All experimental protocols (No. SYXK 2013-0052) were approved by the Animal Care and Use Committee of the Shanghai Jiaotong University. The study took place at the experimental study farm of a feeding company (Xinnong Feed ltd, Shanghai, China) during May and June 2015.

### 2.1. SM, FSM, and SIF Content

The first grade SM was bought from the Yihai Kerry Group (Shanghai, China). The FSBM was prepared by Shanghai Yuanyao Biological Co., Ltd. (Shanghai, China). Briefly, SM and water (SM: Water = 1: 0.85) were mixed for 30 min to produce soybean meal pulp, which was sterilized in a steam tank at a temperature from 80 °C to 90 °C for 30 min. The sterilized SM was naturally cooled down to room temperature, and then the *Aspergillus oryzae* and *Lactobacillus reuteri* were inoculated in the SM, which was fermented in a packed-bed incubator for 48 h. The FSBM pulp was dried at 55 ± 5 °C to moisture at 10 ± 2% and then grounded with a hammer mill to 40 mesh sieve fineness. Samples of SM and FSM were taken and mixed according to the Association of Official Analytical Chemists 2000 (AOAC, 2000) methods [16]. The nutrient components and the content of soy isoflavone in SM and FSBM used in the experiment are shown in Table 1. The total nitrogen (TN) was determined using the macro-Kjeldahl method, and according to (AOAC, 2000) methods [16], crude protein was then calculated as TN × 6.25. Dietary SIFs were determined as described by Barnes et al. [17]. Briefly, SIFs were extracted from soybean meal with 90% methanol solution and detected through HPLC with a 4.6 mm × 150 mm analytical column.

### 2.2. Animal and Experiment Design

A total of 44 sows (Landrace, parity 2 to 4, average parity 3) on the 78th day of gestation were randomly selected and allotted to four dietary groups (*n* = 11 sows/group). The dietary groups included the basal diet group (control) and three treatment groups in which SM was replaced with 2%, 4%, and 6% FSBM in the basal diet (2-FSBM, 4-FSBM, 6-FSBM), respectively. The ingredients and the calculated nutrients levels in the basal diets are shown in Table 2. The standard creep feed is produced by Xinnong Feed ltd (No. JiaoCaoWang, Shanghai, China). The experimental period was from the 78th day of gestation to the 21st day of lactation. The experimental diets were formulated according to the requirements in NRC 2012 for sows [18].

### 2.3. Feeding and Management

All sows at the same round were individually reared in gestational crates in one hog house. On the 110th day of gestation, each sow was transferred to a farrowing crate in two adjacent farrowing rooms. One farrowing room had five replicates of each treatment, and the other farrowing room had six replicates of each treatment. The temperature of the farrowing room ranged from 26 °C to 29 °C. The gestational sows were fed with a 3.0 kg diet per day (supplied two times at 03:30 and 13:30) from the 78th day of gestation to the 110th day of gestation. From the 110th day of gestation to the farrowing day, feed allowance was decreased by 500 g per day until no feed was supplied on the farrowing day. On the second day after farrowing, sows were supplied diet three times per day (06:00, 13:00, and 18:00) with 750 g per day initially, and diet was then increased gradually by 750 g per day until reaching *ad libitum*. Sows had free access to water during the whole experimental period.

All suckling piglets were housed in the corresponding farrowing unit (2.23 m × 2.2 m) with an incubator and heat lamp for piglets from the farrowing day to the weaning day. The room temperature of farrowing units was approximately 26–29 °C. At parturition, the data of litter size and the weight of each new-born piglet were recorded immediately. During 48 h post-farrowing, the litter size was equalized to achieve 10–13 pigs by means of cross-fostering within the same treatment group according to the number of effective nipples in each sow. In addition, the uniformity was adjusted in each litter in the same treatment in order to avoid the relatively small size newborn piglet not getting enough milk before weaning. Piglets per litter were supplied the standard creep feed of 500 g per day (13:00 and 18:00) from the 14th day to the 21st day after birth. Piglets freely had access to water. The litter weight and the litter size were recorded on the 14th day and the 21st day after birth. The piglets were weaned on the 21st day after birth.

### 2.4. Sample Collection

The farrowing process was monitored for sample labeling, data recording, and sample collection. The reproductive performance of sows was evaluated based on the litter size, alive litter size, and litter birth weight of their offspring (before suckling). The survival rate of each litter was recorded at birth. On the 14th day and the 21st day of lactation, nursing piglets were weighed individually. Six sows were randomly selected from each group on the farrowing day and the 17th day of lactation. The blood samples of sows were taken from the vena cava anterior and kept in heparinized tubes. All of the blood samples were centrifuged at 3000× *g* for 15 min, and then the upper serum was collected into 2.0 mL Eppendorf tubes (500 Eppendorf Tubes PCR clean, No,0030123.344, Eppendorf AG, Hamburg, Germany) and stored at −20 °C for further analysis. The colostrum of the same selected sows was obtained (without exogenous oxytocin) within 3 h after the first piglet delivery from the four thoracic pairs of functional mammary glands. Briefly, piglets were separated from their dams for 60 min initially, and the four thoracic pairs of functional mammary glands were milked manually to collect colostrum samples. Approximately 40 mL of colostrum was collected and then separated into two tubes. The colostrum was centrifuged at 4000× *g* at 4 °C for 30 min. The upper fat was removed by using a pipette. The whey of colostrum was collected and then was frozen at −20 °C.

### 2.5. Serum Anti-Oxidative Analysis

The activities of glutathione peroxidase (GSH-Px) (Cat No. A005-1-2) and superoxide dismutase (SOD) (Cat No. A001-1-2), and the levels of malondialdehyde (MDA) (Cat No. A003-1-2) and inhibition of hydroxyl ion (OH−) (Cat No. A018-1-1) in serum were measured according to the instructions of the respective commercial kits (Nanjing Jiancheng Bioengineering Institute, Nanjing, China).

### 2.6. Hormone and Growth Factors

The levels of cortisol (Cat No.H094), 8-iso-prostaglandinF2α (8-iso-PGF2α) (Cat No.H100), estrogen (Cat No.H102), prolactin (PRL) (Cat No.H095), insulin-like growth factor I (IGF-1) (Cat No.H041) and epidermal growth factor (EGF) in serum (Cat No. H031) were determined by using the porcine ELISA assay kits following the protocols of the manufacturer (Nanjing Jiancheng Bioengineering Institute, Nanjing, China). Absorbance values were determined by a microplate reader at 450 nm (Synergy 2, BioTek, Winooski, VT, USA). A four parameters logistic curve-fit was generated using ELISA Calc software v0.1 (Comple-Software. Iowa City, IA, USA). The concentrations of cortisol, 8-iso-PGF2α, estrogen, PRL, IGF-1, and EGF in serum were calculated by comparison with respective standard curves.

### 2.7. Colostrum Composition

The contents of protein, fat, lactose, total solids (TS), and solid not fat (SNF) in colostrum were analyzed through a fully automatic milk analyzer (Milko ScanTM FT+ Analyzer, Foss, Denmark). The contents of Immunoglobulin A (IgA) (Cat No. 42548), immunoglobulin G (IgG) (Cat No. 42547), and immunoglobulin M (IgM) (Cat No. 42546) of colostrum were analyzed using the porcine immunoglobulin ELISA assay kits following the protocols of the manufacturer (Shanghai Yuanye Bio-Technology Co., Ltd., Shanghai, China). Absorbance values were analyzed through a microplate reader (Synergy 2, BioTek, USA). The contents of IgA, IgG, and IgM in milk were calculated by comparison with their respective standard curves.

### 2.8. Statistical Analysis

The individual sow or the litter of piglet were considered as the experimental unit. All variables were tested for normal distribution by Shapiro–Wilk test. The data about blood variance were analyzed by one-way analysis of variance (ANOVA) and multiple comparisons based on the method of Liu et.al (2014) [19]. All the other values in the study were presented as mean ± SEM, and differences among the groups were analyzed by using one-way ANOVA multiple comparisons with LSD post hoc test. A *p*-value less than 0.05 was considered to be statistically significant. All data were analyzed by the software SPSS 17.0 (2012, IBM-SPSS Inc., Chicago, IL, USA).

## 3. Results

### 3.1. Performance of Sows and Piglets

The average weight of piglets was increased in the FSBM groups in comparison with that in the control on the 14th day and the 21st day after birth (*p* < 0.05) (Table 3). The average weight per weaned piglets was increased in the 4-FSBM groups in comparison with that in the control (*p* < 0.05). Replacing SM with FSBM in the maternal diet had no effect on the litter size, alive litter size, the survival rate at birth, litter weight at birth, and the birth weight of piglet (*p* > 0.05). On the 14th day after birth, the weight of litter was heavier in the 4-FSBM and 6-FSBM groups than that of the control (*p* < 0.05), and replacing SM with FSBM in the maternal diet improved the weight of litter in the 2-FSBM, 4-FSBM, and 6-FSBM groups (*p* < 0.001). On the 21st day after birth, the weight of litter was heavier in the 4-FSBM group than that in the control (*p* < 0.05).

### 3.2. Serum Oxidative Stress Status in Sows

Replacing SM with FSBM in the maternal diet decreased the levels of MDA, cortisol, and 8-iso-PGF2α in the serum of sows (Table 4). The inhibition of OH− in the serum of sows was increased on the farrowing day in the 6-FSBM group (*p* < 0.05). The activity of GSH-Px in the serum of sows was higher in the 2-FSBM group than that in the other groups on the farrowing day and the 17th day of lactation (*p* < 0.05), and the activity of SOD in the serum of sows was higher in the 4-FSBM group than that in the control (*p* < 0.05). The levels of cortisol and 8-iso-PGF2α in the serum of sows were lower in both 4-FSBM and 6-FSBM groups than those in the control on the 17th day of lactation (*p* < 0.05).

### 3.3. Serum Hormone and Growth Factors in Sows

Compared with the control group, the level of estrogen in the serum of sows was increased in the 6-FSBM group on the farrowing day (*p* < 0.05) (Table 5). On the 17th day of lactation, replacing SM with 2% FSBM or 6% FSBM increased the level of estrogen in the serum of sows (*p* < 0.05) (Table 5). In addition, the level of PRL in the serum of sows was lower in the 4-FSBM group than that in the other groups (*p* < 0.05).

The level of EGF in the serum of sows was higher in the 4-FSBM and 6-FSBM groups than that in the control on the farrowing day (*p* < 0.05) (Table 5). On the 17th day of lactation, the level of IGF-1 in the serum of sows was greater in the 4-FSBM and 6-FSBM groups than that in the control (*p* < 0.05); the level of EGF in the serum of sows was greater in the 6-FSBM group than that in the control (*p* < 0.05).

### 3.4. Composition of Colostrum

Replacing SM with FSBM in the maternal diet increased the proportion of the protein and TS in the colostrum (*p* < 0.05) (Table 6). The proportions of the protein and TS in the colostrum from the 4-FSBM and 6-FSBM groups were higher than those from the control group (*p* < 0.05), while the proportions of the protein and TS in the colostrum among the FSBM treatment group were not different (*p* > 0.05). The proportions of SNF and the content of IgG in the colostrum from the 4-FSBM group were higher than those from the control group (*p* < 0.05), while SNF and the content of IgG in the colostrum among the FSBM treatment group were not different (*p* > 0.05).

## 4. Discussion

Our results showed that fermentation increased the content of SIFs as the form of aglycones in FSBM, and consequently when fed to sows, attenuated the oxidative stress status of sows during gestation and lactation period, and improved the growth of their offspring.

Consistent with the report of Cho et al. (2011), our data showed that the content of SIFs as the form of aglycones in FSBM significantly increased after fermentation. During the fermentation process, microbial β-glucosidase plays a vital role in the increase of free isoflavone aglycones [6]. The absorption by the body depends on the chemical structures and metabolites of the isoflavones. It has been documented that β-glucosidase hydrolyzes the β-1, 6 glucosidic bonds to increase the concentration of free isoflavone aglycones, which is absorbed easier by the intestinal wall and has greater bioavailability than the glucoside forms [5,20]. Considering the increased biotransformation efficacy of isoflavone aglycones, which have an effect on preventing oxidative stress in animals [21], the FSBM might attenuate the oxidative stress status by increasing the content of SIFs as the form of aglycones.

In addition, our results are in accordance with a previous study, which showed that the supplementation of soybean flavonoid had no effect on the litter size, litter birth weight, and piglet birth weight [22]. However, a different study reported that the soy isoflavones daidzein could enhance the birth weight of male piglets [23]. The varying results might be related to the differences in character, source, feeding period, dose in the diet, and the development stage of the fetus.

During the late gestation and the lactation periods, the high requirement for energy and oxygen favors oxidative stress status in the maternal body because of the excessive production of ROS, such as O^−^, H_2_O_2_, and OH^−^ [24]. These free radicals increase oxidative stress and can produce strong damage to cell structures. In our study, we found that the inhibitory ability of OH^−^ was significantly enhanced after replacing SM with 6% FSBM. This result suggests that FSBM might decrease the oxidative stress in the serum of sows by inhibiting the OH^−^. Furthermore, our results showed that the activity of GSH-Px in the serum of sows was increased in the 2-FSBM group on the farrowing day and the 17th day of lactation, and the activity of SOD in the serum of sows was increased in the 4-FSBM group on the 17th day of lactation. Similarly, on the farrowing day, compared with the control, the 6-FSBM group had a significantly increased level of estrogens in the serum of sows. The estrogen exerts antioxidant effect via unexpected regulatory roles on oxidative stress [25,26]. Moreover, during the process of lipid peroxidation, oxidative stress can be caused by the production of the MDA and 8-iso-PGF2α, which are the specific marker of oxidative stress [27,28]. The flavonoid aglycones were more potent in their antiperoxidative action than their corresponding glycosides [29]. In our study, the levels of MDA and 8-iso-PGF2α in the serum of sows were decreased in the FSBM groups on the farrowing day and the 17th day of lactation. Similarly, the level of cortisol indicates the status of a stress response [30]. Our results showed that the cortisol in the serum of sows was decreased in all FSBM groups on the farrowing day and in the 4-FSBM and the 6-FSBM groups on the 17th day of lactation. These results suggest that replacing SM with FSBM in maternal diets might reduce oxidative stress status and enhance the anti-oxidative capacity of sows on the farrowing day and the 17th day of lactation via increasing the anti-oxidative products or reducing the oxidative stress products.

The breast milk components include fat, fat-soluble vitamins and minerals, immunoglobulins, and specific fatty acids [12]. The weight gain of piglets is closely linked to the milk components in the swine industry because of colostrum and milk supply nutrients and the immune agents for the new-born piglets [12,31]. The composition of colostrum has positive impacts on the levels of antioxidants [32,33]. A previous study has shown that antioxidants can increase the cysteine (Cys) and glutamine (Glu) of whey protein (WP) [34]. In our study, we found that the concentrations of TS, SNF, and IgG in the colostrum were significantly increased by replacing SM with 4% FSBM in the maternal diet. These results might be linked to not only the increase of the SIFs as the form of aglycones but also the decrease of oxidative stress of sows during the lactation period. The improvement of milk components could be beneficial to the growth performance of piglets.

In addition, our study about the growth performance of the piglets is consistent with the study of Kim et al., who suggested that the starter feed of calf with the FSBM enhanced the body weight gain and the feed intakes [15]. The reasons for the increased average daily gain of piglets might be related to the milk composition and immunoglobulin levels, which are key factors for the growth of nursing piglets [35,36]. The milk production of the mammary epithelial cells was regulated by the hormones, such as PRL, IGF-1, and EGF [37]. The gene expression of milk proteins was regulated by EGF in the presence of PRL in ex vivo mammary gland cultures [38], but the dose-response relationship between PRL and EGF in regulating the secretion of milk protein is still unclear. Interestingly, our results showed that the level of PRL in the serum of sows was decreased, but the level of EGF in the serum of sows had no significant change in the 4-FSBM group on the 17th day of lactation; the level of PRL in the serum of sows had no significant change, but the level of EGF in the serum of sows was increased in the 6-FSBM group on the 17th day of lactation. Further studies are needed to determine the levels of EGF and PRL in the mammary gland of sows and the relationship between the hormones and the milk production in the mammary gland.

EGF is beneficial in protecting neonates against pathogen infection by enhancing neonatal intestinal development [39,40]. IGF-1 might be beneficial for improving neonatal tissue development [41]. In our study, replacing SM in the maternal diet with FSBM increased the levels of IGF-1 and EGF in the serum of lactation sows and the average weight of piglets on the 14th day and the 21st day after birth. These results suggested that replacing SM in the maternal diet with FSBM might enhance the growth and development of the suckling piglets after birth. Further studies are needed to determine the levels of EGF and IGF-1 in the milk of sows and the serum of their offspring in different lactating periods for analyzing the connection between the growth and health of the piglets and the hormones.

## 5. Conclusions

FSBM increased the content of SIFs in the form of aglycones, and thus, the maternal diet with FSBM attenuated the oxidative stress status in sows. Replacing SM in the maternal diet with FSBM attenuated the oxidative stress status of the gestational and lactational sows, improved the milk components, and further enhanced the average daily gain of suckling piglets after birth, but the dose-dependent effect was not observed in the study. The mechanism of the fermented soybean meal to regulate oxidative stress merits further investigation.

## Figures and Tables

**Table 1 animals-11-00597-t001:** Crude protein and soy isoflavone content of the soybean meal (SM) and fermented soybean meal (FSBM) used in the experimental diets.

Items	SM	FSBM	*p* Value
Crude protein (%)	44.72 ± 1.18	47.88 ± 1.42	NS
Isoflavone glucoside			
Underivatized glucosides (μg/g)	575.12 ± 17.43	100.33 ± 4.56	*p* < 0.001
Acetyl glucosides conjugates (μg/g)	139.67 ± 10.43	22.70 ± 1.38	*p* < 0.001
Malonyl glucosides (μg/g)	872.71 ± 4.07	168.07 ± 6.14	*p* < 0.001
Isoflavone aglucone			
Daidzein (μg/g)	157.18 ± 4.82	392.37 ± 15.43	*p* < 0.001
Glycitin (μg/g)	55.49 ± 2.27	130.34 ± 5.27	*p* < 0.001
Genistein (μg/g)	155.76 ± 10.99	1210.87 ± 34.42	*p* < 0.001

SM = soybean meal; FSBM = fermented soybean meal; NS = not significant. Values are means ± SEM, *n* = 4.

**Table 2 animals-11-00597-t002:** Dietary ingredients and nutrient contents of the basal diets (as fed-basis).

Ingredient	Day 78–84 of Gestation				Day 85–100 of Gestation				Day 100 of Gestation Lactation			
Corn (%)	33.9	33.9	33.9	33.9	34.8	34.8	34.8	34.8	54.0	54.0	54.0	54.0
Barley (%)	44.0	44.0	44.0	44.0	35.0	35.0	35.0	35.0	12.3	12.3	12.3	12.3
Wheat bran (%)	5.00	5.00	5.00	5.00	-	-	-	-	-	-	-	-
SM (%)	13.1	11.1	9.10	7.10	11.2	9.20	7.20	5.20	13.5	11.5	9.50	7.50
FSBM (%)		2.00	4.00	6.00		2.00	4.00	6.00		2.00	4.00	6.00
Extruded soybean (%)	-	-	-	-	12.0	12.0	12.0	12.0	10.0	10.0	10.0	10.0
NUPRO ^1^ (%)	-	-	-	-	-	-	-	-	2.50	2.50	2.50	2.50
Fish meal (%)	-	-	-	-	-	-	-	-	1.00	1.00	1.00	1.00
HTL-306 ^1^ (%)	-	-	-	-	3.00	3.00	3.00	3.00	2.50	2.50	2.50	2.50
L-Lysine-HCL,98% (%)	-	-	-	-	-	-	-	-	0.10	0.10	0.10	0.10
DL-methionine, 99% (%)	-	-	-	-	-	-	-	-	0.05	0.05	0.05	0.05
L-Threonine, 98.5% (%)	-	-	-	-	0.05	0.05	0.05	0.05	0.05	0.05	0.05	0.05
Gestation-premix ^2^ (%)	4.00	4.00	4.00	4.00	4.00	4.00	4.00	4.00	-	-	-	-
Lactation-premix ^3^ (%)	-	-	-	-	-	-	-	-	4.00	4.00	4.00	4.00
Total (%)	100	100	100	100	100	100	100	100	100	100	100	100
Calculated nutrients levels												
Metabolizable energy, kcal/kg	3029	3032	3035	3038	3208	3211	3214	3217	3306	3309	3312	3315
Crude protein, %	14.2	14.3	14.4	14.5	15.9	16.0	16.1	16.2	16.7	16.8	16.9	17.0
Ether extract,%	2.49	2.46	2.42	2.39	7.30	7.26	7.23	7.19	6.97	6.84	6.81	6.77
Crude fiber,%	3.99	3.88	3.78	3.67	3.75	3.65	3.54	3.44	3.17	3.06	2.96	2.85
Lysine, %	0.65	0.66	0.66	0.66	0.80	0.80	0.80	0.80	1.08	1.08	1.08	1.08
Total calcium, %	0.88	0.88	0.88	0.88	0.90	0.90	0.90	0.90	0.96	0.96	0.96	0.96
Total phosphorus, %	0.54	0.55	0.55	0.55	0.52	0.53	0.53	0.54	0.52	0.52	0.53	0.53

^1^ HTL-306: fat powder, purchase from Berg + Schmidt, Hamburg, Germany; NUPRO: yeast extract, purchase from Alltech Biological Products CO., LTD. (Beijing, China). ^2^ Gestation-premix provided per kilogram of diets: vitamin A, 10450 IU; vitamin E, 60 IU; vitamin K3, 5.5 IU; vitamin B1, 2.0 mg; vitamin B2, 6.0 mg; vitamin B6, 3.0 mg; vitamin B12, 25 μg; folate, 1.5 mg; biotin, 250 μg; pantothenic acid, 25.2 mg; niacin, 35.4 mg; choline, 400 mg; Ca, 6.61 g; P, 2.34 g; Cu, 35 mg; Fe, 200 mg; Mn, 30.5 mg, Zn, 150 mg; Cr, 0.15 mg; Se, 0.30 mg; I, 0.6 mg. ^3^ Lactation-premix provided per kilogram of diets: Lys, 0.78 g; vitamin A, 6890 IU; vitamin D3, 2950 IU; vitamin E, 60 IU; vitamin K3, 6.0 IU; vitamin B1, 2.0 mg; vitamin B2, 6.0 mg; vitamin B6, 3.0 mg; vitamin B12, 25 μg; folate, 1.5 mg; biotin, 300 μg; pantothenic acid, 25.2 mg; niacin, 35.4 mg; choline, 400 mg; Ca, 7.94 g; P, 2.16 g; Cu, 35 mg; Fe, 250 mg; Mn, 30.5 mg, Zn, 150 mg; Cr, 0.15 mg; Se, 0.30 mg; I, 0.6 mg.

**Table 3 animals-11-00597-t003:** The performance of sows and their litter in different dietary groups.

Items	Control	2-FSBM	4-FSBM	6-FSBM
Litter size	12.00 ± 0.98	11.55 ± 0.87	12.64 ± 0.77	11.09 ± 0.71
Alive litter size	11.27 ± 0.83	10.82 ± 0.72	12.00 ± 0.79	10.45 ± 0.64
Survival rate at birth	0.95 ± 0.01	0.94 ± 0.02	0.95 ± 0.02	0.95 ± 0.02
The number of weaned piglets per litter	10.91 ± 0.39	10.45 ± 0.51	10.18 ± 0.18	10.18 ± 0.54
Average daily gain per litter (kg)	1.58 ± 0.19 ^a^	1.91 ± 0.13 ^a^	2.04 ± 0.16 ^b^	1.93 ± 0.17 ^a^
Weight/litter (kg)				
Birth	16.98 ± 0.77	16.76 ± 0.94	18.06 ± 0.85	16.17 ± 1.74
14 day old	34.78 ± 2.22 ^a^	40.43 ± 2.23 ^a,b^	44.18 ± 1.37 ^b^	41.96 ± 2.87 ^b^
21 day old	50.15 ± 3.95 ^a^	57.01 ± 2.90 ^a,b^	60.95 ± 1.91 ^b^	56.78 ± 3.59 ^a,b^
Weight/piglets, kg				
Birth	1.57 ± 0.11	1.58 ± 0.88	1.54 ± 0.07	1.55 ± 0.11
14 day old	3.17 ± 0.13 ^a^	3.91 ± 0.20 ^b^	4.37 ± 0.19 ^b^	4.16 ± 0.25 ^b^
21 day old	4.57 ± 0.30 ^a^	5.53 ± 0.29 ^b^	6.03 ± 0.28 ^b^	5.65 ± 0.34 ^b^

Control = soybean meal-based diet (*n* = 11); 2-FSBM = soybean meal-based diet replaced soybean meal with 2% FSBM (*n* = 11); 4-FSBM = soybean meal-based diet replaced soybean meal with 4% FSBM (*n* = 11); 6-FSBM = soybean meal-based diet replaced soybean meal with 6% FSBM (*n* = 11). ^a,b^ Mean values within a row with unlike superscript letters differ significantly (*p* < 0.05).

**Table 4 animals-11-00597-t004:** The oxidative stress status in the serum of sows in different dietary groups.

Items	Control	2-FSBM	4-FSBM	6-FSBM
The 1st day of lactation				
GSH-Px ^1^ (U/mL)	357.61 ± 10.9 1 ^a^	426.18 ± 13.19 ^b^	373.68 ± 17.04 ^a^	374.74 ± 17.04 ^a^
SOD ^2^ (U/mL)	49.14 ± 0.65	48.26 ± 2.36	50.35 ± 1.23	48.89 ± 0.85
MDA ^3^ (nmol/mL)	9.89 ± 0.44 ^a^	6.25 ± 0.29 ^b^	6.44 ± 0.42 ^b^	6.94 ± 0.71 ^b^
inhibition of OH− ^4^ (U/mL)	213.06 ± 5.26 ^a^	214.12 ± 4.29 ^a^	222.66 ± 3.80 ^a,b^	231.69 ± 4.10 ^b^
8-ISO-PGF2α ^5^ (pg/mL)	92.07 ± 2.17 ^a^	85.00 ± 1.27 ^b^	82.61 ± 2.00 ^b^	86.19 ± 1.10 ^b^
The 17th day of lactation				
GSH-Px ^1^ (U/mL)	428.28 ± 19.00 ^a^	479.02 ± 17.2 ^b^	449.68 ± 12.67 ^a,b^	443.12 ± 9.28 ^a,b^
SOD ^2^ (U/mL)	41.95 ± 2.89 ^a^	45.78 ± 1.45 ^a,b^	49.22 ± 1.11 ^b^	45.35 ± 1.48 ^a,b^
MDA ^3^ (nmol/mL)	7.52 ± 0.15 ^a^	4.02 ± 0.46 ^b^	4.66 ± 0.28 ^b^	6.37 ± 0.34 ^c^
inhibition of OH− ^4^ (U/mL)	160.77 ± 4.10 ^a^	163.60 ± 3.03 ^a,b^	174.15 ± 3.72 ^b^	205.76 ± 3.45 ^c^
8-ISO-PGF2α ^5^ (pg/mL)	103.36 ± 3.88 ^a^	100.15 ± 4.00 ^a^	41.91 ± 2.57 ^b^	72.64 ± 1.83 ^c^

Control = soybean meal-based diet (*n* = 6); 2-FSBM = soybean meal-based diet replaced soybean meal with 2% FSBM (*n* = 6); 4-FSBM = soybean meal-based diet replaced soybean meal with 4% FSBM (*n* = 6); 6-FSBM = soybean meal-based diet replaced soybean meal with 6% FSBM (*n* = 6). ^1^ GSH-Px = glutathione peroxidase; ^2^ SOD = superoxide dismutase; ^3^ MDA = malondialdehyde; ^4^ OH− = hydroxyl ion; ^5^ 8-ISO-PGF2α=8-iso-prostaglandin F2α; ^a,b,c^ Mean values within a row with unlike superscript letters differ significantly (*p* < 0.05).

**Table 5 animals-11-00597-t005:** The levels of estrogen, prolactin, IGF-1, and EGF in the serum of sows.

Items	Control	2-FSBM	4-FSBM	6-FSBM
The 1st day of lactation				
Estrogen (pg/mL)	74.96 ± 1.87 ^a^	75.59 ± 0.32 ^a^	76.27 ± 2.77 ^a^	88.05 ± 1.86 ^b^
Prolactin (ng/mL)	36.37 ± 2.54	36.96 ± 2.96	39.38 ± 1.78	38.96 ± 2.00
IGF-1 ^1^ (ng/mL)	5.95 ± 0.11	6.02 ± 0.34	6.55 ± 0.11	6.51 ± 0.26
EGF ^2^ (ng/L)	204.32 ± 9.19 ^a^	199.01 ± 5.54 ^a^	226.08 ± 6.56 ^b^	231.3 ± 6.39 ^b^
Cortisol (ng/mL)	87.12 ± 2.00 ^a^	80.73 ± 1.89 ^b^	71.22 ± 1.60 ^c^	75.86 ± 2.64 ^b,c^
The 17th day of lactation				
Estrogen (pg/mL)	69.65 ± 1.94 ^a^	76.50 ± 1.81 ^b,c^	71.61 ± 2.73 ^a,b^	79.18 ± 1.78 ^c^
Prolactin (ng/mL)	48.41 ± 2.19 ^a^	51.02 ± 2.47 ^a^	40.79 ± 3.06 ^b^	51.01 ± 2.31 ^a^
IGF-1 ^1^ (ng/mL)	6.33 ± 0.145 ^a^	6.49 ± 0.18 ^a,b^	6.95 ± 0.18 ^b,c^	7.05 ± 0.19 ^c^
EGF ^2^ (ng/L)	225.37 ± 9.27 ^a^	214.95 ± 13.37 ^a^	243.51 ± 11.82 ^a,b^	269.85 ± 5.61 ^b^
Cortisol (ng/mL)	75.96 ± 0.50 ^a^	74.66 ± 1.14 ^a^	66.76 ± 1.24 ^b^	70.45 ± 1.73 ^c^

^a,b,c^ Mean values within a row with unlike superscript letters differ significantly (*p* < 0.05). Control = soybean meal-based diet (*n* = 6); 2-FSBM = soybean meal-based diet replaced soybean meal with 2% FSBM (*n* = 6); 4-FSBM = soybean meal-based diet replaced soybean meal with 4% FSBM (*n* = 6); 6-FSBM = soybean meal-based diet replaced soybean meal with 6% FSBM (*n* = 6).^1^ IGF-1 = insulin-like growth factor; ^2^ EGF = epidermal growth factor. Composition of Colostrum.

**Table 6 animals-11-00597-t006:** The composition and immunoglobulin levels of colostrum from different dietary groups.

Items	Control	2-FSBM	4-FSBM	6-FSBM
Fat (%)	3.90 ± 0.13	4.79 ± 0.40	4.50 ± 0.99	4.73 ± 0.37
Protein (%)	13.56 ± 0.66 ^a^	15.08 ± 0.53 ^a,b^	16.76 ± 0.83 ^b^	17.59 ± 1.39 ^b^
Lactose (%)	2.52 ± 0.38	2.74 ± 0.10	2.48 ± 0.28	2.19 ± 0.31
TS ^1^ (%)	22.49 ± 0.70 ^a^	25.39 ± 0.85 ^a,b^	26.67 ± 1.23 ^b^	27.7 ± 1.31 ^b^
SNF ^2^ (%)	17.73 ± 0.78 ^a^	21.30 ± 1.59 ^a,b^	22.48 ± 1.30 ^b^	21.82 ± 1.26 ^a,b^
IgA ^3^ (μg/mL)	3.33 ± 0.15	3.36 ± 0.18	3.67 ± 0.15	3.74 ± 0.16
IgG ^3^ (mg/mL)	2.09 ± 0.03 ^a^	2.19 ± 0.06 ^a^	2.45 ± 0.12 ^b^	2.30 ± 0.08 ^a,b^
IgM ^3^ (μg/mL)	13.45 ± 0.35	14.06 ± 0.40	14.17 ± 0.28	14.03 ± 0.34

Control = soybean meal-based diet (*n* = 6); 2-FSBM = soybean meal-based diet replaced soybean meal with 2% FSBM (*n* = 6); 4-FSBM = soybean meal-based diet replaced soybean meal with 4% FSBM (*n* = 6); 6-FSBM = soybean meal-based diet replaced soybean meal with 6% FSBM (*n* = 6). ^1^ TS = total slid; ^2^ SNF = solid not fat; ^3^ IgA = immunoglobulin A; IgG = immunoglobulin G; IgM = immunoglobulin M; ^a,b^ Mean values within a row with unlike superscript letters differ significantly (*p* < 0.05).

## Data Availability

The data used to support the findings of this study are included within the article file.

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
