# Peer review of "Fermented Soybean Meal Affects the Reproductive Performance and Oxidative Status of Sows, and the Growth of Piglets"

_animals, 2021, doi:10.3390/ani11030597_

Round 1
Reviewer 1 Report
Dear authors,
You modified the manuscript according to suggestions, but I think that there are some important mistakes in Table results:
- Table 3. survival rate is not on a percentage basis.
- Table 4: MDA concentration must be expressed per ml of serum?,
- Table 5: unit for IGF-I is still wrong in the second row.
- Headings for days relative of farrowing unfitted...
These mistakes should have detected by co-authors during the draft preparation, and hinder a trustful view of the blood results...that in fact are difficult to compare with other references (rare markers?). Did you compare your absolute values with other literature references?
In blood variables it would be necessary to use mixed models with repeated measurements so that to take into account individual variability across sows and be trust of the outcomes, as some differences may be irrelevant when using this statistical method. In addition, cortisol was placed as an oxidative marker in table 4 but it is in fact a hormone more related with those of table 5.
Author Response
[Comments1]
1. Table 3. survival rate is not on a percentage basis.
2. Table 4: MDA concentration must be expressed per ml of serum?,
3. Table 5: unit for IGF-I is still wrong in the second row.
4. Headings for days relative of farrowing unfitted...
These mistakes should have detected by co-authors during the draft preparation, and hinder a trustful view of the blood results...that in fact are difficult to compare with other references (rare markers?). Did you compare your absolute values with other literature references?
[Answer 1]Thank the reviewer for the valuable comments. We are so sorry for these mistakes in these tables and apologize for our careless. We had deleted the unit in the Table 3 of the revised manuscript and had revised the unit in the table 4 and table 5 of the revised manuscript. We have replaced “the day of farrowing” into “the 1st day of lactation” in the table 4 and table 5 of the revised manuscript. In addition, we have compared the level of several reference about the pigs’ markers in our manuscript. The index of oxidative status and the hormone about the milk secretion in serum affected by the breed, diet, environment, condition of animals, age, tissue and the management in the farm, so the absolute date might be a wide range. We had listed some of these range according to the reference as in table 1. Our result are accordance to the range for these markers items. Furthermore, the determined method and the manufacture of the kits might have effect on the absolute value of the items.
[Comments2]In blood variables it would be necessary to use mixed models with repeated measurements so that to take into account individual variability across sows and be trust of the outcomes, as some differences may be irrelevant when using this statistical method. In addition, cortisol was placed as an oxidative marker in table 4 but it is in fact a hormone more related with those of table 5.
[Answer 2] Thank the reviewer for the valuable comments. We have moved the cortisol data in the table 5 of the revised manuscript according to your comments. Statically, the mixed models with repeated measurements analyze the blood variables might be get outcomes with different differences. We analyzed these data with one-way ANNOVA and LDS post hoc test according to the previous study. The previous study analyzed the sows’ plasma biochemical parameters and immunoglobulin levels on day 107 of gestation, day of farrowing and day of weaning (day 24 of lactation) through the one-way analysis of variance (ANOVA) and multiple comparisons to get the results[10]. We have supplemented the statistical method reference in the statistical analysis part of the revised manuscript.
[1] Lv, M., et al., Responses of growth performance and tryptophan metabolism to oxidative stress induced by diquat in weaned pigs. Animal, 2012. 6(6): p. 928-34.doi:10.1017/S1751731111002382.
[2] Zhu, L., et al., Effect of N-acetyl cysteine on enterocyte apoptosis and intracellular signalling pathways' response to oxidative stress in weaned piglets. Br. J. Nutr., 2013. 110(11): p. 1938-47.doi:10.1017/S0007114513001608.
[3] Xu, J., et al., Regulation of an antioxidant blend on intestinal redox status and major microbiota in early weaned piglets. Nutrition, 2014. 30(5): p. 584-9.doi:10.1016/j.nut.2013.10.018.
[4] Zhen Luo, W.Z., Qi Guo,Wenli Luo, Jing Zhang, Weina Xu, and Jianxiong Xu, Weaning Induced Hepatic Oxidative Stress,Apoptosis, and Aminotransferases through MAPK Signaling Pathways in Piglets. 2016.
[5] Pan, D.-S., et al., Plasma 8-iso-Prostaglandin F2α, a possible prognostic marker in aneurysmal subarachnoid hemorrhage. Clin. Chim. Acta, 2017. 469: p. 166-170.doi:https://doi.org/10.1016/j.cca.2017.04.005.
[6] Hendricks, D., Estrogen concentrations in bovine and porcine tissues. J. Toxicol. Environ. Health, 1976. 1: p. 617-39.doi:10.1080/15287397609529362.
[7] Farmer, C., Altering prolactin concentrations in sows. Domest. Anim. Endocrinol., 2016. 56: p. S155-S164.doi:https://doi.org/10.1016/j.domaniend.2015.11.005.
[8] Burrin, D.G., et al., Orally administered IGF-I increases intestinal mucosal growth in formula-fed neonatal pigs. American Journal of Physiology-Regulatory, Integrative and Comparative Physiology, 1996. 270(5): p. R1085-R1091.doi:10.1152/ajpregu.1996.270.5.R1085.
[9] Roh, S.-G., J.A. Carroll, and S.W. Kim, Effects of fermented soybean meal on innate immunity-related gene expressions in nursery pigs acutely challenged with lipopolysaccharides. Anim Sci J, 2015. 86(5): p. 508-516.doi:10.1111/asj.12319.
[10] Liu, S.T., et al., Effects of dietary citric acid on performance, digestibility of calcium and phosphorus, milk composition and immunoglobulin in sows during late gestation and lactation. Anim. Feed Sci. Technol., 2014. 191: p. 67-75.doi:10.1016/j.anifeedsci.2014.01.017.
Reviewer 2 Report
The authors made the requested corrections. However, minor errors occur in the corrected text. These should be improved.
line86: It is mentioned twice in the sentence: "according to"
line 89: macro kjeldahi method, correctly: macro-Kjeldahl method
line 133: Please specify the standard creep feed manufacturer (product name, producer)
Author Response
[Comments]
The authors made the requested corrections. However, minor errors occur in the corrected text. These should be improved.
[Comment 1]line86: It is mentioned twice in the sentence: "according to"
[Answer 1] Thank the reviewer for the valuable suggestion. We have deleted one of “according to“ in the Line 86 of the revised manuscript.
[Comment 2]line 89: macro kjeldahi method, correctly: macro-Kjeldahl method.
[Answer 2] Thank the reviewer for the valuable comments.We have replaced the ‘macro kjeldahi method’ into ‘macro-Kjeldahl method’ in the Line 88 of the revised manuscript.
[Comment 3]line 133: Please specify the standard creep feed manufacturer (product name, producer)
[Answer 3] Thank the reviewer for the valuable comments.We have specified the standard creep feed manufacturer in the Line100-101 of the revised manuscript.
This manuscript is a resubmission of an earlier submission. The following is a list of the peer review reports and author responses from that submission.
Round 1
Reviewer 1 Report
There are some formal errors in the text that need to be corrected.
line 17: average body weight of piglets
line 50-52: This sentence must be corrected (hih efficacy)
line 88: Kjeldahl method
line 103: calculated nutrient contents
Please add to the text
line 96: average body weight of sows (missing value)
line 103: what kind of bran? wheat bran?
line 103: please specify the amino acids! L-Lysine-HCL, DL-methionine, etc. with amino acid content too.
line 103: the metabolizable energy content of the diets (missing data)
line 175: mean±sd?
line 176: with LSD post hoc test?
line 180: more data on reproductive performance of sows suggested to add to this chapter
line 227: Chapter 3.4. Please give more detailed analysis of the data!
line 315: This part (Conclusions) must be rewritten. I suggest that lines 305-314 should be moved to the conclusion chapter. Line 318 is OK in the present form.
Reviewer 2 Report
The objective of the work was to examine the effect of increasing amount of fermented SBM during late gestation and lactation on sow and litter performance. The paper is well written with just a few changes needed.
Line 102 The NRC 2012 should be included in the cited references.
Line 103 Table 2 has a few errors. Instead of “Nutrition levels” it would be better to use “Calculated nutrients”
There are no values for metabolizable energy.
Line 116 What is meant by “at the same round”? Clarify if all 44 sows were in the same farrowing group or if the study was conducted over several groups.
Line 123 Should be “free access”
Line 130 This is unusual management. If pigs stayed in the farrowing crate with the sow until day 21, they would continue to suckle and eat creep feed. Clarify what is meant by “pigs only suckled the milk of sows from day 0-14?
Line 143 Were the sows that had blood samples collected the same as the ones that had colostrum collected?
Line 145 Why was colostrum centrifuged? It would seem more appropriate to analyze the complete colostrum. Since milk fat would be on the surface of centrifuged samples, it is not clear how it was defatted.
Line 172 I do not understand why this was not analyzed for the linear and quadratic effects of fermented SBM. This would seem to be a more appropriate analysis for this type experiment. Doing this would allow you to create tables with the LS Means for each treatment, an SEM and then the P-values for Linear and Quadratic effects. This would make it much easier to see responses to treatment.
Line 189 The number of pigs weaned per litter should be stated. It is not clear how many pigs were cross fostered. I think this should be stated in the results. Also pre-weaning mortality should be stated.
Also, was sow body weight monitored?
Line 210 Was return to estrus monitored in the sows? I realize that the numbers of animals are small, but it might be of interest to know if this was changed.
The objective of the work was to examine the effect of increasing amount of fermented SBM during late gestation and lactation on sow and litter performance. The paper is well written with just a few changes needed.
Line 102 The NRC 2012 should be included in the cited references.
Line 103 Table 2 has a few errors. Instead of “Nutrition levels” it would be better to use “Calculated nutrients”
There are no values for metabolizable energy.
Line 116 What is meant by “at the same round”? Clarify if all 44 sows were in the same farrowing group or if the study was conducted over several groups.
Line 123 Should be “free access”
Line 130 This is unusual management. If pigs stayed in the farrowing crate with the sow until day 21, they would continue to suckle and eat creep feed. Clarify what is meant by “pigs only suckled the milk of sows from day 0-14?
Line 143 Were the sows that had blood samples collected the same as the ones that had colostrum collected?
Line 145 Why was colostrum centrifuged? It would seem more appropriate to analyze the complete colostrum. Since milk fat would be on the surface of centrifuged samples, it is not clear how it was defatted.
Line 172 I do not understand why this was not analyzed for the linear and quadratic effects of fermented SBM. This would seem to be a more appropriate analysis for this type experiment. Doing this would allow you to create tables with the LS Means for each treatment, an SEM and then the P-values for Linear and Quadratic effects. This would make it much easier to see responses to treatment.
Line 189 The number of pigs weaned per litter should be stated. It is not clear how many pigs were cross fostered. I think this should be stated in the results. Also pre-weaning mortality should be stated.
Also, was sow body weight monitored?
Line 210 Was return to estrus monitored in the sows? I realize that the numbers of animals are small, but it might be of interest to know if this was changed.
The objective of the work was to examine the effect of increasing amount of fermented SBM during late gestation and lactation on sow and litter performance. The paper is well written with just a few changes needed.
Line 102 The NRC 2012 should be included in the cited references.
Line 103 Table 2 has a few errors. Instead of “Nutrition levels” it would be better to use “Calculated nutrients”
There are no values for metabolizable energy.
Line 116 What is meant by “at the same round”? Clarify if all 44 sows were in the same farrowing group or if the study was conducted over several groups.
Line 123 Should be “free access”
Line 130 This is unusual management. If pigs stayed in the farrowing crate with the sow until day 21, they would continue to suckle and eat creep feed. Clarify what is meant by “pigs only suckled the milk of sows from day 0-14?
Line 143 Were the sows that had blood samples collected the same as the ones that had colostrum collected?
Line 145 Why was colostrum centrifuged? It would seem more appropriate to analyze the complete colostrum. Since milk fat would be on the surface of centrifuged samples, it is not clear how it was defatted.
Line 172 I do not understand why this was not analyzed for the linear and quadratic effects of fermented SBM. This would seem to be a more appropriate analysis for this type experiment. Doing this would allow you to create tables with the LS Means for each treatment, an SEM and then the P-values for Linear and Quadratic effects. This would make it much easier to see responses to treatment.
Line 189 The number of pigs weaned per litter should be stated. It is not clear how many pigs were cross fostered. I think this should be stated in the results. Also pre-weaning mortality should be stated.
Also, was sow body weight monitored?
Line 210 Was return to estrus monitored in the sows? I realize that the numbers of animals are small, but it might be of interest to know if this was changed.
The objective of the work was to examine the effect of increasing amount of fermented SBM during late gestation and lactation on sow and litter performance. The paper is well written with just a few changes needed.
Line 102 The NRC 2012 should be included in the cited references.
Line 103 Table 2 has a few errors. Instead of “Nutrition levels” it would be better to use “Calculated nutrients”
There are no values for metabolizable energy.
Line 116 What is meant by “at the same round”? Clarify if all 44 sows were in the same farrowing group or if the study was conducted over several groups.
Line 123 Should be “free access”
Line 130 This is unusual management. If pigs stayed in the farrowing crate with the sow until day 21, they would continue to suckle and eat creep feed. Clarify what is meant by “pigs only suckled the milk of sows from day 0-14?
Line 143 Were the sows that had blood samples collected the same as the ones that had colostrum collected?
Line 145 Why was colostrum centrifuged? It would seem more appropriate to analyze the complete colostrum. Since milk fat would be on the surface of centrifuged samples, it is not clear how it was defatted.
Line 172 I do not understand why this was not analyzed for the linear and quadratic effects of fermented SBM. This would seem to be a more appropriate analysis for this type experiment. Doing this would allow you to create tables with the LS Means for each treatment, an SEM and then the P-values for Linear and Quadratic effects. This would make it much easier to see responses to treatment.
Line 189 The number of pigs weaned per litter should be stated. It is not clear how many pigs were cross fostered. I think this should be stated in the results. Also pre-weaning mortality should be stated.
Also, was sow body weight monitored?
Line 210 Was return to estrus monitored in the sows? I realize that the numbers of animals are small, but it might be of interest to know if this was changed.
Reviewer 3 Report
This study investigated the effect of replacing soyabean meal (SBM) with fermented SBM in the maternal diets on anti-oxidative status, reproductive performance, colostrum composition of peripartum sows, and the performance of their offspring.
The topic is of interest, but most of earlier works are focused on the use of fermented SBM in weaned piglets to improve their digestive welfare. However, this ingredient may not affordable in other production phases as in reproductive sows (high cost! economic return?).
The main drawback is that I wonder if oxidative stress was really a problem in peripartum sows farrowing 12 piglets, as in the present study? It seems unfeasible, unless body condition of the sows was extremely low.
When googling the reader can find other recent works dealing with similar topic, although they were not referenced herein:
Wang, C., Lin, C., Su, W., Zhang, Y., Wang, F., Wang, Y.,Shi, C., Lu, Z. 2018. Effects of supplementing sow diets with fermented corn and soybean meal mixed feed during lactation on the performance of sows and progeny. Journal of Animal Science, 96(1), 206-214.https://doi.org/10.1093/jas/skx019
Other specific comments:
Table 1. Amino acid composition of fermented SBM compared to SM would be required (as this ingredient have effect on milk protein content). In addition, the reader wonders how you collect so many feed samples to analyse their compositional differences statistically (not detailed earlier)
Table 2. Crude fibre seems to be reduced by including fermented SBM. Is this ingredient rich in oligosaccharides? Some other dietary fibre measurement would be required to support digestive and metabolic welfare.
Table 3. You should provide the ADG of piglets during lactation to support differences in milk production (or hormonal mediators in sows)
L115 how did you assure the individual feed intake during gestation?
L144 how could you collect colostrum without oxytocin supply? Could you fill 40 ml from four thoracic glands? Please detail procedure to allow replication
Table 4, the SOD content on the day of farrowing in 6-FSBM is not correct, please revise.
SOD and IGF-I are expressed as U/ml, but in the case of IGF-I, can this unit be used for hormones? what does it mean really?
More details in discussion may be valuable to understand the role of 8-ISO-PGF2α in nutrition.